# An Analysis of the Spatiotemporal Characteristics and Diversity of Grain Production Resource Utilization Efficiency under the Constraint of Carbon Emissions: Evidence from Major Grain-Producing Areas in China

**DOI:** 10.3390/ijerph19137746

**Published:** 2022-06-24

**Authors:** Haokun Wang, Hong Chen, Tuyen Thi Tran, Shuai Qin

**Affiliations:** 1School of Economics and Management, Northeast Forestry University, Harbin 150040, China; whk@nefu.edu.cn (H.W.); tranthi_tuyen@yahoo.com (T.T.T.); 15225145321@nefu.edu.cn (S.Q.); 2Ecological Civilization Construction and Green Development Think Tank of Heilongjiang Province, Harbin 150040, China

**Keywords:** grain production, three-stage super-efficiency EBM, major producing areas, external environmental factors, utilization efficiency

## Abstract

As the most important driving force for ensuring the effective supply of grain in the country, the production stability of the major grain-producing areas directly concerns the national security of China. In this paper, considering the “water–soil–energy–carbon” correlation, water, soil and energy resource factors, and carbon emission constraints were included in an index system, and the global common frontier boundary three-stage super-efficient EBM–GML model was used to measure the grain production resource utilization efficiency of the major grain-producing areas in China from 2000 to 2019. This paper also analyzed the static and dynamic spatiotemporal characteristics and the restrictions of utilization efficiency. The results showed that, under the measurement of the traditional data envelopment analysis model, the grain production resource utilization efficiency in the major producing areas is relatively high, but there is still room to improve by more than 20%, and grain production still has enormous growth potential. After excluding external environmental and random factors, it was found that the utilization efficiency of grain production resources in the major producing areas decreased, and the efficiency and ranking of provinces changed significantly. External factors inhibit pure technical efficiency and expand the scale efficiency. The utilization efficiency of Northeast China was much higher than that of the Huang-Huai-Hai region and the middle and upper reaches of the Yangtze River region, and its grain production resource allocation management had obvious advantages. The total factor productivity index of food production resources showed an upward trend as a whole, and its change was affected by both technological efficiency and technological progress, of which technological progress had the greater impact. Therefore, reducing the differences in the external environment of different regions while making adjustments in accordance with their own potential is an effective way to further improve the utilization efficiency of food production resources.

## 1. Introduction

Since ancient times, food has always been a top priority in China’s governance and national security. As a country with a large population and a relative shortage of water and soil resources, China uses merely 7% of the world’s total freshwater resources and 8% of the world’s arable land resources to produce 1/4 of the food and feed about 1/5 of the population of the world. This is no mean feat. In other words, resolving China’s food problems is equivalent to resolving the world’s food security difficulties [1]. It has been over 70 years since the founding of the People’s Republic of China. Over the past seven decades, grain output has increased nearly fivefold, and the per capita grain supply has doubled. In addition, serving as the main driving force that guarantees effective national grain supplies, the major producing areas have long been responsible for more than 75% of the national grain output and more than 90% of the increase in output. However, the negative externalities brought about by the “Three High” mode of agriculture, such as tightening restraints on resources, low efficiency of inputs, and ecological environment degradation, have become intractable problems for increasing grain production, which seriously discourages the development of the food industry in China. The resource utilization efficiency of grain production is an important index with which to measure the input–output relationship in production. Guided by green development, we should implement the rural revitalization strategy and the national food security strategy. It is necessary to adjust and optimize the existing production resource elements. We need to achieve the goal of increasing grain production by improving the efficiency of resource allocation. Therefore, in order to scientifically and accurately measure the utilization efficiency of grain production resources in major grain-producing areas, we should explain its characteristics and identify the existing problems. We need to devise effective ways of improving grain productivity and to further develop the food security theory.

Resource allocation is a hot issue in the field of food or agricultural security, which is the reason why numerous agricultural resources and environmental studies are based on this. There are many choices in the methods of measuring resource utilization efficiency. These methods include the life cycle method, the ratio analysis method, the index system method, the production function method (C-D), stochastic frontier analysis (SFA), and data envelopment analysis (DEA). Most studies focus on agricultural allocative efficiency. For example, Reddy [2] used Malmquist productivity indices (MI) to examine regional differences in agricultural productivity growth in Andhra Pradesh from 1956 to 2007 and found that there was convergence in total factor productivity growth between developed coastal areas and less developed areas. Total irrigated area, fertilizer use, and labor supply were the limiting factors for increasing yield. Additionally, market infrastructure and the availability of credit are critical for improving efficiency. Ferreira and Feres [3] adopted the stochastic production frontier model to measure farm scale and land use efficiency in the Amazon region of Brazil in order to explore their relationship. They found that land use intensification still has room to expand and that there was a u-shaped relationship between these two factors. Toma et al. [4] used data envelopment analysis to measure agricultural efficiency in EU countries from 1993 to 2013. It was found that all EU countries have experienced the process of increasing or decreasing economies of scale. It was pointed out that European countries have the potential to improve production efficiency by adjusting inputs, and the reason for efficiency improvement is the implementation of the common agricultural policy. Most studies in the literature involve the stochastic forward edge method and the data envelopment analysis method [5]. As a parameter method, the stochastic frontier analysis method sets the production function based on economic theory and identifies calculation errors to facilitate statistical inference. Studies on agricultural allocative efficiency are mainly carried out using efficiency measurements and their influencing factors [6,7], spatial effect [8,9], and coordinated economic development [10]. However, data envelopment analysis has become the mainstream method in agricultural allocative efficiency research because it is a non-parametric method and has many advantages, such as avoiding constructing production functions, the dimension of input–output index has no influence, there is no subjective empowerment, and there are improved relaxation variables. Most researchers use the DEA method to measure allocative efficiency evaluation [11,12,13] and identify influencing factors and spatial effects [14,15,16]. 

It can be found that, although the conventional DEA model can solve the practical differences in efficiency in the process of agricultural production, the evaluation of efficiency in each region does not consider the impact of external environmental factors. In particular, the real agricultural resource allocation level, production and utilization efficiency, management level evaluation, and other aspects are unfair. Therefore, a few scholars have tried to explore a more comprehensive and accurate efficiency evaluation method by applying the three-stage DEA model proposed by Fried et al. [17,18]. The model is a combination of the data envelopment analysis method and the stochastic frontier production function method. While retaining the advantages of the non-parametric method, the influences of environmental factors and random error on the efficiency of decision unit are fully considered. This method makes up for the disadvantages of the previous DEA evaluation of being interfered with by external factors. On this basis, some researchers have studied agricultural productivity in different years or regions. Among them, Guo et al. [19] and He and Liu [20] evaluated and classified China’s agricultural production efficiency in 2008 and 2012, respectively. Pan et al. [21] used macro data to analyze the changes in agricultural production efficiency in 11 provinces of the Yangtze River Economic Belt over a decade. A few researchers have analyzed the agricultural production efficiency of the adopters of conservation tillage using micro-survey data and discussed the influence of environmental factors on agricultural production efficiency [22]. In addition, some scholars have refined the second stage and analyzed the difference in agricultural production efficiency from the perspectives of the environmental effect and luck difference of efficiency [23,24]. Wang et al. [25] pointed out that the difference in external environment was due to different agricultural policies. Chen et al. [24] and Zhang et al. [26] conducted internal regional evaluations of the production efficiency from functional grain areas and major producing areas during the same period. However, it was found that the conclusions about the changing direction of efficiency of major producing areas and the influence of the external environment are contradictory. It can be found that studies on agricultural allocation efficiency do not consider regional heterogeneity, and the same categories can be compared to each other as the basic premise of DEA measurement. In particular, the agricultural production objectives, endowments, and characteristics of different regions are not the same, and the measurement results analyzed in the unified model may not accurately reflect reality [27,28]. Additionally, the choice of input and output index building is relatively constant, mainly in production input elements (labor, land, fertilizers, pesticides, machinery, etc.) or the idea of constructing total factor productivity. Most studies have failed to take into account the large statistical deviations of agricultural labor force indicators in different regions. With the rapid development of urbanization in China, farmers have become part-time farmers, and most of them are not engaged in agricultural production [29,30,31]. Therefore, it is difficult to determine the size of the labor force truly participating in agricultural production from the panel data of the National Bureau of Statistics; thus, it is better to reflect on the accurate situation of the resource allocation level of grain production by excluding labor force indicators in index construction. Water, soil, energy, and carbon are important resource elements in the agricultural production process, which have long received due academic attention [32,33,34,35]. Some of them are studied from the perspective of a single factor, for example, carbon emissions, water footprints, arable land use, and energy analysis [36,37,38,39]. A few studies have been carried out in terms of two or more factors, for example, from the perspective of soil and water resource loss [40], utilization and matching [33,41], or expansion from the perspective of carbon emission of water and soil [42,43]. Additionally, some have considered the water, soil, and carbon correlation perspective to study agricultural efficiency [44,45,46,47]. Therefore, on the basis of previous research results, it is worth further analyzing the utilization efficiency of food production resources considering the water–soil–energy–carbon relationship as an indicator.

To sum up, it is a mature and effective method to study the allocation efficiency of grain production resources using a data envelopment analysis. However, in the design of input–output indicators, the separation of environmental factors and random errors is still difficult in academic circles but important for forming the basis of policy for the design of agricultural resource allocation in major grain-producing areas. Therefore, this paper considered water, soil, energy, and carbon as important input–output indicators in the process of grain production and applied the global common frontier three-stage super-efficiency EBM–GML model to scientifically measure the utilization efficiency of grain production resources in major grain-producing areas in China from 2000 to 2019 to identify the impact of the external environment on efficiency. The research significance of this paper lies in identifying the impact of various external environments on grain resource allocation efficiency, finding out the intensity and direction of the impact, evaluating the resource allocation efficiency level of each main grain producing area, analyzing its dynamic and static spatiotemporal characteristics and constraints, and theoretically analyzing the potential of grain production in each region. It provides policy reference for the improvement of grain production efficiency in major grain producing areas.

## 2. Overview of the Study Area, Research Methods, and Data Selection

### 2.1. Overview of the Study Area

According to the proposals of the State Council on deepening the reform of the grain circulation system in 2001, 31 provinces (autonomous regions and municipalities directly under the Central Government) were divided into 13 major producing areas, 7 major marketing areas, and 11 balanced production and marketing areas. Among the major grain-producing areas (green parts in Figure 1), Heilongjiang, Jilin, Neimenggu, Henan, and Anhui are net grain-transferring provinces, while Jiangsu, Shandong, Jiangxi, Hunan, and Hebei are rich in grain production and demand. The other three provinces, Liaoning, Hubei, and Sichuan, have relatively low grain production and demand. With the continuous advancement of industrialization and urbanization, the former Beidahuang (the Great Northern Wilderness in Northern China) has now become the northern warehouse, with abundant grain, and the transportation of grain from the south to the north has also been transformed into the other way round. As the major grain supply base in China, its strategic significance is self-evident. According to the Statistical Yearbook of China, China’s total grain output in 2021 was 682.85 million tons, and it has remained above 650 million tons for seven consecutive years. The grain output of the 13 major grain-producing areas was 536.03 million tons, accounting for 78.55% of the total, and the unit yield was about 5621 kg per hectare. In particular, the five largest grain-producing areas accounted for 99% of the country’s grain transfers. As China’s demand for food continues to rise, the amount of organic matter in arable land is only 43% of the world average. In such an environment, China has begun to pay attention to green and low-carbon development, especially as the pure fertilization intensity decreased from 370 kg/ha in 2015 to 313 kg/ha in 2021. It can be said that stable grain production in major grain-producing areas directly concerns national food security. Therefore, it is of great value to research the major grain-producing areas. Due to the relativity of the envelope analysis method and the homogeneity required to enable its application, this paper selected 13 provinces in the major producing areas as the research subject. 

### 2.2. The Research Methods

Data envelopment analysis is an efficiency evaluation and analysis method based on the principle of linear programming that is described by the distance function. The measurement result was based on the relative efficiency value of the input and output, reflecting whether the resource allocation can achieve the relative optimal state. With the continuous development of the data envelopment analysis method in theoretical research and practical applications, dozens of improved methods have emerged. Among them, Tone and Tsutsui [48,49] proposed the super-efficient SBM model by combining the SBM model with the super-efficient method to realize the optimization of non-expected value, exclude the decision unit from the participation, and remove the constraint that the efficiency value of effective decision unit is below 1, but it is still difficult to avoid the loss of the original proportion information of the projected value of the efficiency frontier. In this regard, Tone and Tsutsui [50] proposed a mixed EBM model containing radial and SBM distance functions. The GML index is also known as the total factor productivity index with global reference, which avoids the defects of non-transitivity and linearity existing in the traditional ML index by constructing a global production possibility set [51]. As a large number of studies have found that heterogeneity is not considered in DEA models, common frontier analysis DEA models are gradually proposed and applied. The super-efficiency EBM model can eliminate the evaluated DMU from the reference set, so the efficiency value of the evaluated decision-making unit (DMU) is obtained by referring to the front surface formed by other DMU so that the effective DMU value is not limited by 1 so that the effective DMU can be distinguished. The GML index was developed on the basis of the Malmquist–Luenberger (ML) index. This method can effectively avoid the problem of the ML index having no linear solution, and the GML index has the characteristics of transitivity and multiplicativity. The GML index can be decomposed into the efficiency change (GEFC) and technology change (GETC). GEFC can be further decomposed into the pure efficiency change (GPEC) and scale efficiency change (GSEC), and GETC can be further decomposed into the pure technology change (GPTC) and technology scale change (GSTC).

Therefore, the EBM–GML model can avoid failing to estimate the efficiency loss because the time of research and DMU are in the same production frontier. In terms of the agricultural DEA efficiency study, part of the factors contributing to the improvement of agricultural production efficiency include the roles of external environment and random interference, which is not entirely caused by the improvement of technological level and production scale. The three-stage DEA model can identify the influences of various external environments on the evaluation of agricultural efficiency, including the external differences of agricultural policy, agricultural investment, and agricultural natural endowment. By eliminating these external differences in agricultural production, the production efficiency of each region or farmer can be reasonably evaluated theoretically, and the technical level of agricultural resource allocation can be reflected. Therefore, based on previous studies and the above analysis, the author put forward the research hypothesis of this paper according to the relationship between the allocation efficiency of food production resources, relaxation variables, and the external environment. Research hypothesis H1: The external environment significantly affects the relaxation variables of various food production resources. Research hypothesis H2: The difference in external environment reduces the overall allocation efficiency of food production resources in major grain-producing areas. Research hypothesis H3: Under the influence of the external environment, the direction and intensity of grain production resource allocation efficiency in different regions are different. Based on the characteristics of the above model, this paper constructed the global common frontier boundary three-stage super-efficient EBM–GML model. The main purpose was to combine the common foreword super-efficiency EBM model with the SFA model and to use the external environmental factors to strip the slack variables of the input index and obtain all the slack variables caused by the management factors. The processed variables are re-calculated to obtain more accurate relative effective values. At present, this paradigm method has become relatively common for measuring and breaking down the dynamic changes in TFP. It can effectively avoid the lack of clearly vertical comparison benchmarks due to the differences in production fronts during different periods. The method is as follows:

The first stage and the third stage: Phase one and phase three had the same approach. The first stage was the traditional DEA model. It used the principle of mathematical programming to obtain efficiency according to multiple groups of input–output data. The total efficiency value obtained was the product of the allot efficiency and technical efficiency. Subsequently, the assumption of fixed return to scale was changed to variable return to scale by the scholars′ constant revision so that the technical efficiency was decomposed into scale efficiency and pure technical efficiency; technical efficiency = scale efficiency × pure technical efficiency, total efficiency = technical efficiency × allot efficiency. In this way, the low scale and low efficiency of production technology were identified, and the two reasons for the low efficiency of technology were also found. 

The second stage: Using the method proposed by Fried et al. [17,18], in this stage, the *N* independent stochastic frontier regression was estimated, and all input relaxation variables caused by the management factors were obtained by regression stripping of the input relaxation variables in the first stage. Furthermore, the processed variables were put back into the super-efficiency EBM–GML model for accurate measurement. The specific formula is as follows:(1)$$\begin{array}{c} Sxy=f\left(z_{yi};\beta_{yi}\right)+v_{xy}+\mu_{xy}\\ \left(x=1,2,\cdots ,X;y=1,2,\cdots ,Y;i=1,2,\cdots ,I\right) \end{array}$$
where *S_xy_* in Formula (1) is the relaxation variable of the input index *y* of decision unit *x* after the calculation of the super-efficiency EBM before adjustment. *z_mi_* is the external environmental variable affecting efficiency change, and *β_yi_* is the coefficient of *z_mi_*. *v_xy_* and *μ_xy_* are the mixed bias terms, where *v_xy_* represents the random disturbance factors affecting the efficiency change, and *μ_xy_* is the management factor affecting the relaxation variable. Both the random disturbance factors and the management factors follow the normal distribution with a mean value of 0 and a variance of $\sigma_{v}^{2}$. *β_mi_*, *v_nm_*, and *μ_nm_* were derived according to the separation of *μ_xy_* by its predecessors, and the parameters were substituted into Formula (2) to further adjust the input index values:(2)$$\begin{array}{c} X_{nm}^{*}=X_{nm}+\left[f{\left(z_{mi};\beta_{mi}\right)}^{*}+v_{nm}^{*}\right]\\ \left(n=1,2,\cdots ,N;m=1,2,\cdots ,M;i=1,2,\cdots ,I\right) \end{array}$$

In Formula (2), $X_{mn}^{*}\;$ refers to the input quantity after removing environmental and random factors by means of stochastic frontier analysis. *F* (*z_mi_; β_mi_*) is the maximum, and max is the subtraction of *f*( *z_mi_; β_mi_*) and *f*( *z_mi_; β_mi_*).$\;v_{mn}^{*}$ is the maximum and max is the subtraction of $\;v_{mn}^{*}$ and $v_{mn}^{*}$.

In the third stage, the revised input index value was re-invested into the super-efficiency EBM–GML model for measurement, and the relatively real efficiency value that is almost immune from external influences was obtained. The first stage represents the actual allocative efficiency of grain production (utilization efficiency), and the third stage reflects the resource allocation and management level of grain production in each region after the adjustment of the second stage.

### 2.3. Variables Selection and Data Sources

#### 2.3.1. Input–Output Indicator

This paper was based on the panel data of 13 major grain-producing areas in China from 2000 to 2019. The term ‘grain’ in this paper means a general term for four major crops including rice, wheat, corn, and soybean. In terms of index selection, referring to the existing research, it considers the water-soil-energy-carbon correlation, the grain water footprint (water), crop planting area (soil), chemical fertilizer applied (chemical), and total power machinery (mechanical) as input indicators. Among them, the data of fertilizer and total power machinery need to be converted according to the proportion of the sown area of the four grains in the sown area of agriculture. The water footprint of crops was calculated separately for the four crops and then added together (see Appendix A). In terms of the output index, in order to offset the impact of price changes on the grain output value, the total output of the four major crops was taken as the desired output, and the total carbon emissions (carbon) in the converted grain production process were taken as the undesired output.

#### 2.3.2. External Environmental Indicators

External factors affecting the utilization efficiency of food production resources mainly fall into two categories: external environmental factors and random errors. As external environmental factors are non-direct input factors in efficiency measurements and are not controlled by decision-making units, some random errors can also be eliminated when external environmental factors are eliminated using the stochastic frontier analysis model. This paper referred to the research of Pan et al. [21] and Chen et al. [52]. Based on data availability, this paper selected the urbanization level, economic development level, affected degree, resource endowment, and agricultural financial support as indicators of environmental variables, which are represented by urbanization rate, per capita GDP, affected area, per capita arable land area, and the local fiscal spending on agriculture, forestry, and water as specific indicators, respectively.

#### 2.3.3. Data Source

The research methods of Xu and Mu [53] and Cao et al. [54] were used to calculate the water footprint of grain production in this paper. The crop growth data and meteorological data were derived from the “study on the water demand contour map of China’s major crops (books)”, “Cropwat Software”, the database of The United Nations Food and Agriculture Organization, the China Meteorological Data Network (http://data.cma.cn, (accessed on 15 October 2021)), provincial statistical yearbooks, and previous research [52,55]. The calculation of carbon emissions from grain production adopted the coefficients and methods used by Cheng et al. [56] and Li Bo et al. [57], and the basic data of carbon sources were collected from provincial statistical yearbooks. In addition to the calculation of crops’ water footprint and carbon emissions, the remaining data indicators and variables were derived from provincial statistical yearbooks, the National Bureau of Statistics database (https://data.stats.gov.cn, (accessed on 8 January 2022)), and the “China rural statistical yearbook” from 2001 to 2020.

## 3. Results 

### 3.1. Empirical Analysis of the First Stage of Traditional Super-Efficient EBM

In this section, by applying the MaxDEA 8 Ultra software measurement and the input-oriented common frontier super-efficiency EBM model, considering the undesired outputs, the grain production resource utilization efficiency of 13 major grain-producing areas in China from 2000 to 2019 was estimated. The comprehensive technical efficiency shown in this paper is a comprehensive measurement and evaluation of the allocation capacity or utilization efficiency of grain production resources in each region under the assumption of constant scale. In this paper, the technical efficiency is the technical level of effective matching of grain production resource elements under the assumption of variable returns to scale, while scale efficiency is the efficiency separated from both, reflecting the scale of grain production resource allocation. The specific efficiency measurement results of the first stage are shown in Table 1.

Table 1 shows the measurement results of the traditional envelope analysis model. Without considering various areas under the influence of external environmental factors and random errors, Jilin, Jiangxi, and Heilongjiang provinces have the highest efficiency level of grain resource allocation, while Shandong, Hebei, and Anhui are the lowest-level provinces. From 2000 to 2019, the grain production resource allocation efficiency in major producing areas varied greatly, with a weighted average of 0.733 but a total distance value of 0.361. The weighted average of the pure technical efficiency in each region was 0.767, and the differences between the pure technical efficiency in each region were also significant, with the gap between extreme values being 0.352. The distribution of the pure technical efficiency was similar to that of the comprehensive technical efficiency. The weighted average of scale efficiency in each region was the highest, reaching 0.957, and the median was as high as 0.977. Compared with pure technical efficiency and comprehensive technical efficiency, the gap among extreme values was relatively low, merely 0.120. Comprehensive technical efficiency and pure technical efficiency increased as a whole, with comprehensive technical efficiency increasing from 0.727 to 0.798 and the pure technical efficiency increasing from 0.782 to 0.871. However, after 2014, affected by the decline in scale, the synergy degree of the two decreased year by year. As a result, the growth rate of pure technical efficiency was much higher than that of comprehensive technical efficiency. Meanwhile, scale efficiency tended to be flat from 2000 to 2014 and declined slightly year by year from 2015, decreasing from 0.938 to 0.927. In conclusion, without considering the influence of external environmental factors and random errors in major grain-producing areas, the scale efficiency of major grain-producing areas in China is relatively high on the whole, and the improvement of comprehensive technical efficiency is mainly restricted by the level of pure technical efficiency.

### 3.2. SFA Regression Results of the Second Stage

In order to feasibly compare the utilization efficiency of grain production resources in major grain-producing areas, it is necessary to clarify the influence degree of external factors on the utilization efficiency of grain production resources in each region and strip out the influence of the external environment and random errors on the utilization efficiency of grain production resources. Therefore, in the second stage, stochastic frontier analysis software (FRONTIER Version 4.1 (The computer program FRONTIER Version 4.1 was written by Tim Coelli who is professor at the Centre for Efficiency and Productivity Analysis (CEPA) in Australia.)) was used to conduct SFA regression estimation. In the regression model, taking the relaxation variables of input indicators in the first stage as dependent variables and taking urbanization rate, GDP per capita, agricultural disaster area, arable land per capita area, and local government expenditure on agriculture, forestry, and water conservancy as independent variables, if the regression coefficient is positive and significant, an increase in the environmental variable will cause an increase in the slack variables, and the efficiency of food production resources is thus hindered (see Appendix A). The results are shown in Table 2.

The SFA regression results in Table 2 show that the Log Likelihood function value and single-tailed error test value (LR) achieved better estimation effects, and the model was set reasonably. Most of the regression coefficients can pass the significance test, and the gamma value of each input relaxation variable was close to 1, indicating that, in the mixed error term, management factors play a dominant role in the influence of each input relaxation variable, and random factors have little influence on the input relaxation variable, so it is necessary to exclude the environmental and random factors.

(1)The regression of the relaxation variables of the urbanization level on the amount of fertilizer application and the total power of machinery passed the significance test of 1%. The regression coefficients of the relaxation variables of the total power of chemical fertilizer machinery were negative and positive, respectively, which indicates that the increasing urbanization level will reduce the redundancy of the chemical fertilizer application amount in the grain production process and increase the redundancy of the agricultural capital investment, mainly based on the total power resources of mechanical machinery.(2)The regression coefficients of the three relaxation variables of the degree of disaster on the water footprint of grain production and the sowing area of grain and the amount of fertilizer application were all negative, and all of them passed the significance test of at least 5%. The greater the degree of disaster, the lower the relaxation variable, indicating that disasters have more of an impact on low-level farmland, which is consistent with the conclusions of He and Liu et al. [20] and Zhang et al. [58]. In the wake of a natural disaster, modern agricultural technologies are able to provide support for farmland disaster recovery. In addition to the effective replacement of resources, the advanced natural disaster prevention and control system and the construction of high-standard farmland have consolidated China’s ability to ensure food security so that it can effectively resist the influence of natural disasters on agricultural production.(3)The regression of economic development to the four input relaxation variables all passed the significance test of 1%. Among them, the coefficients of water footprint of grain production, sowing areas of grain, and total mechanical power were all negative, and the relaxation variable of fertilizer application was positive. This means that greater economic development in each region in the major grain-producing areas can effectively reduce the redundancy of the three inputs and, on the other hand, will further increase the redundant input of chemical fertilizer application, which reflects actual agricultural production.(4)The regression of resource endowment on the relaxation variables of grain sowing areas and fertilizer application amount passed the significance test at 1%. Resource endowment had a positive impact on the input redundancy of the grain sowing area, but it had a negative impact on the input redundancy of fertilizer application amount. It indirectly suggested that, with the stimulus of grain increase policy, the more abundant the cultivated land resources are, the more obvious the abuse of chemical fertilizer is, which is also consistent with the current situation that the increase in grain output in China mainly depends on the excessive input of chemical fertilizer.(5)The regression coefficients of the relaxation variables of agricultural financial support to the water footprint of grain production and the amount of chemical fertilizer application were positive, and both passed the significance test of at least 5%. The regression results proved that the support policy would increase the input of water resources and the amount of chemical fertilizer application in grain production, resulting in increasing redundancy.

The analysis of the above SFA regression results shows that the external environmental factors affecting the utilization efficiency of food production resources are real and have a significant impact. Therefore, it is necessary to further exclude the impact of environmental factors on efficiency to obtain a real, comparable utilization efficiency of food production resources.

### 3.3. Empirical Analysis of Adjusted Third Stage Super-Efficiency EBM

The initial input variables were adjusted according to the analysis results of the second stage, and the adjusted new input variables were brought into the common frontier super-efficiency EBM model again to obtain the utilization efficiency of food production resources in each region in the third stage (see Appendix A). The specific efficiency values and changes are shown in Table 3 and Figure 2 and Figure 3. After comparing the first stage and the third stage, it was found that, after excluding external environmental factors and random errors, the three indicators of resource utilization efficiency of grain production in major producing areas in China had changed greatly from 2000 to 2019, indicating the necessity of the rational adjustment of input indicators.

In terms of the overall characteristics of comparison, before and after excluding the environmental variables and random factors, the weighted average of comprehensive technical efficiency and scale efficiency of grain production resource allocation efficiency in the major grain-producing areas in China from 2000 to 2019 during the study period decreased as a whole. The comprehensive technical efficiency decreased from 0.733 to 0.639, the scale efficiency decreased from 0.957 to 0.785, and the weighted average of pure technical efficiency increased from 0.767 to 0.822. The changes in the three indicators indicate that the external factors inhibited the pure technical efficiency and expanded the scale efficiency, and finally, the comprehensive technical efficiency was exaggerated with scale efficiency in place. As is shown in Figure 4 and Table 4, the changes in the comprehensive technical efficiency gap before and after adjustment during the study period consisted of two periods, with 2017 as the turning point. From 2000 to 2017, the overall trend of comprehensive technical efficiency before and after adjustment showed synergistic changes, and efficiency gradually increased with time. The efficiency difference decreased from 0.20 to 0.02, and the efficiency gap shows an expanding trend for two consecutive years thereafter. The main reason was the decrease in scale efficiency, especially the scale efficiency after adjustment. Similarly, we can see from the efficiency trend in the figure that external environmental factors were more important in influencing the scale effect of food production resource allocation.

In terms of the spatiotemporal characteristics after adjustment, from 2000 to 2019, the evolution of grain production resource allocation efficiency in China’s major grain-producing areas showed the characteristics of the two periods mentioned above, in which the comprehensive technical efficiency increased from 0.528 to 0.761 with an annual growth rate of 1.94%, the pure technical efficiency increased from 0.828 to 0.911 with an annual growth rate of 0.51%, and the scale efficiency increased from 0.660 to 0.836 with an annual growth rate of 1.26%. The overall data show that the improvement rate of grain production technology was not as fast as that of production scale, and the improvement of utilization efficiency of grain production resources in the major producing areas over the past 20 years mainly depended on the expansion of scale. By observing the changing trend of the three types of efficiency in Figure 4, it can be found that pure technical efficiency fluctuated gently from 2000 to 2009, gradually rose from 2009 to 2017, and then declined continuously from 2018 to 2019. By looking at the scale efficiency, it can be seen that before 2009 the improvement in comprehensive technology efficiency was mainly driven by scale efficiency, and from 2009 to 2017 it was driven by scale efficiency and pure technical efficiency. Meanwhile, from 2018 to 2019, it was mainly driven by pure technical efficiency. In terms of regional distribution (Table 3 and Figure 5), the weighted average of grain production resource utilization efficiency (comprehensive technical efficiency) in the major producing areas was 0.639, and the regions above the average were Heilongjiang, Jilin, Hunan, and Sichuan (in this order), demonstrating that only four provinces have high comprehensive management levels of resource allocation. The weighted mean value of pure technical efficiency was 0.822, and the regions above the mean value were Heilongjiang, Jilin, Jiangxi, Liaoning, Hunan, Neimenggu, Sichuan, and Hubei. More than half of the regions improved their resource allocation purely using technology. The weighted mean value of scale efficiency was 0.785, and the regions above the mean value were Shandong, Henan, Heilongjiang, Hebei, Jiangsu, Anhui, and Jilin, in that order. Likewise, more than half of the regions improved their levels of resource allocation by expanding the scale.

In order to further explore the distribution of the utilization efficiency of grain production resources in different regions, this paper, instead of dividing the research region into eastern, central, and western regions according to the previous economic belt classification standard, combined the characteristics of grain cultivation, natural geographical characteristics, and allocative efficiency level characteristics (the range of dotted lines in Figure 6). Major grain-producing areas were divided into northeast China, the Huang-Huai-Hai region, and the middle and upper reaches of the Yangtze River. The detailed variation trend of the allocation efficiency of food production resources in these three regions is shown in Table 5 and Figure 7. The northeast includes Heilongjiang, Jilin, Liaoning, and Neimenggu; the Huang-Huai-Hai region includes Hebei, Henan, Shandong, Anhui, and Jiangsu; and the middle and upper reaches of the Yangtze River include Jiangxi, Hubei, Hunan, and Sichuan.

### 3.4. Dynamic Analysis of Resource Utilization Efficiency of Grain Production Based on Malmquist Productivity Index

In order to further explore the dynamic changes in production resource utilization efficiency in the major grain-producing areas in China from 2000 to 2019, the GML index model was included in this section, with the aim of showing the changes in resource utilization efficiency of food production by applying total factor productivity. In this section, while measuring the GML index before and after adjustment, it can be decomposed into the technological efficiency change index and technological progress change index to identify the influence degree of both on the dynamic changes in total factor productivity of grain production resources. The former measures the catch-up degree of each decision-making unit to the production frontier border for two consecutive terms, presenting the catch-up effect of resource use efficiency, and it can be further decomposed into pure technical efficiency change and scale efficiency change index change, while the latter measures the mobile cases of the effective frontier during two consecutive periods, presenting the growth effect of resource utilization efficiency. The change index results of decomposition efficiency are shown in Table 6 and Figure 8.

As can be seen in Table 6, on the whole, the utilization efficiency of grain production resources in major producing areas showed a significant upward trend from 2000 to 2019, with each efficiency change index being greater than 1. In particular, the difference degree of the GML change index before and after adjustment was merely 1.59%, indicating that external factors have a low impact on the improvement of total factor productivity, reflecting the sustainability of the continuous improvements in resource utilization efficiency in China’s major grain-producing areas. GML change is influenced by both technological efficiency and technological progress, and technological progress has a greater impact on Malmquist change, indicating that the increase in the total factor productivity of food production resources not only contributes to the catch-up effect but also to the growth effect, and the growth effect takes the initiative. In particular, after excluding external factors, the total factor productivity of resources in northeast China witnessed the fastest increase during the study period, with an average annual growth rate of 4.04%, while that of the middle and upper reaches of the Yangtze River and of the Huang-Huai-Hai region was 1.91% and 0.90%, respectively. In Heilongjiang, Jiangsu, and Hubei, the GML change index was still greater than 1, while Hunan’s technical efficiency declined, indicating that technological progress has a significant effect on the growth of total factor productivity of resources in these four provinces.

It can also be seen from Figure 8 that the GML change index of all years from 2000 to 2019 was greater than 1 except for 2003, 2007, 2009, 2014, and 2018, indicating the overall progress of total factor productivity of food production resources. The GML change is basically consistent with the changing trend of technological progress, which proves once again that the change in the total factor productivity of food production resources mainly depends on technological changes. The GML change index and its decomposition can be divided into the following three stages in terms of trends. The first stage is the synergy stage (2001–2005), where the changing trend of technical efficiency and total factor productivity was coordinated, especially in 2003, the change index of technical efficiency and technological progress were both less than 1, and the catch-up effect and growth effect showed a simultaneous decline. The second stage is the interactive stage (2006–2011), where the GML change index was influenced by the technological progress change index and technological progress change index interactively, and the synergy was not high. In particular, the technological efficiency and technological progress change index varied most dramatically in 2017 and 2018 during the study period. The third stage is the stable stage, in which the technological efficiency and technological progress change index showed a stable fluctuation trend from 2012 to 2019.

## 4. Discussion

The purpose of this study was to explore the allocation efficiency of production resources in major grain-producing areas in China from the perspective of water–soil–energy–carbon linkage, while excluding the impact of the external environment and to identify the driving factors for the improvement of resource allocation efficiency in different regions through analyses of differences. In the selection of its indicators, this study considered that the defect of Chinese farmer quantity indicators would have a negative influence on the results of the data envelopment analysis. In the research method, the homogeneity of the research area was fully taken into account, and the standard analysis paradigm was adopted to avoid the results being unable to reflect reality. The method of this paper was first to establish carbon emission constraints under each grain production factor evaluation system and to analyze the traditional methods of regional resource use efficiency when it comes to grain production. Second, we calculated the external environmental impact on efficiency. Finally, we calculated the efficiency after stripping out the influence of the external environment and evaluated the management level of production resource allocation in each main grain-producing area. 

It was found that, under the measurement of the traditional envelope analysis model, the overall utilization efficiency of grain production resources in the main producing areas was high, but there was still more than 20% room for improvement. After excluding the external environmental factors and random factors, the utilization efficiency of grain production resources in the main grain-producing areas decreased, and the external factors inhibited the pure technical efficiency while expanding the scale efficiency, indicating that there is more room to improve the resource allocation and management level in the main grain-producing areas. In terms of the comparison of spatial features, the direction and intensity of the influence of external factors on the utilization efficiency of food production resources in different regions were obviously different. Except for Hebei and Anhui, which remain unchanged in terms of their efficiency rankings, the regions with rising comprehensive technical efficiency ranking included Heilongjiang (from 3 to 1), Shandong (from 10 to 6), Henan (from 3 to 1), Jiangsu (from 9 to 7), Sichuan (from 5 to 4), and Hunan (from 4 to 3). The regions that fell in the overall technical efficiency rankings include Liaoning (from 7 to 10), Jilin (from 1 to 2), Neimenggu (from 6 to 12), Hubei (from 8 to 9), and Jiangxi (from 2 to 8). Through the comparison of comprehensive technical efficiency values in Table 3, Figure 3 and Figure 5, it can be found that the mean change in grain production resource utilization efficiency in each region in the major grain-producing areas was significantly different before and after adjustment. With the exception of Heilongjiang, Hebei, Shandong, Anhui, Henan, and Jiangsu, efficiency fluctuated greatly in other regions. It was further indicated that external factors have an obvious promotional effect on grain production resource utilization efficiency in Liaoning, Jilin, Sichuan, Hubei, Hunan, Jiangxi, and Neimenggu. Among them, Jiangxi (0.275), Neimenggu (0.272), and Liaoning (0.241) had the greatest promotional effect. After adjustment, except for Shandong and Henan, the average utilization efficiency of grain production resources in other provinces and regions was lower than that before adjustment, indicating that the relative inefficiency of Shandong and Henan before adjustment was caused by an unfavorable environment and random error, rather than its low level of resource allocation and management. In contrast, the previous high efficiency of the other regions was due to a more favorable environment and random errors, thus creating the illusion of high-level resource allocation management. In addition, the differences in the direction of influence also indicated that the external factors do not all have a positive promotional effect on the comprehensive technical efficiency, but they also have a restraining effect, which shows that the external environment of the major grain-producing areas, mainly the comprehensive endowment of grain and policy support, is obviously different. 

The utilization efficiency of Northeast China is much higher than that of the Huang-Huai-Hai region and the middle and upper reaches of the Yangtze River region, and its grain production resource allocation management has obvious advantages. According to the adjusted distribution of pure technical efficiency and scale efficiency in the scattered chart (Figure 6), the 13 major grain-producing areas can be divided into four types. The first type is the two-high type, which includes Heilongjiang, Jilin, Jiangxi, and Liaoning. Its efficiency in both indicators was relatively high, so the improvement room is relatively limited. The second type is the high-and-low type, including Jiangxi, Liaoning, and Neimenggu. Due to its relatively low-scale efficiency, there is greater room for improvement. The utilization efficiency of grain production resources can be improved by enhancing the scale. The third type is the low-and-high type, which includes the Shandong, Henan, Hebei, Jiangsu, and Anhui provinces. Due to their relatively low pure technical efficiency, the specific path to improve the utilization efficiency of food production resources is to improve the management and technical factors. The fourth type is the two-low type, such as Hubei province, where pure technical efficiency and scale efficiency are both relatively low. In theory, it has the largest improvement room, but this area of food production is often restricted by some objective factors. They should adjust their measures to local conditions so as to expand production, adopt new technology, and strengthen management, which will serve as comprehensive governance measures to promote the allocation of food production resources. Combined with Table 4, after adjustment, the average efficiency in the Huang-Huai-Hai region, northeast China, and the middle and upper reaches of the Yangtze River are 0.563, 0.472, and 0.504, respectively. Given the evolving trend of the utilization efficiency of grain production resources in the three regions (Figure 7), since 2005, the major grain-producing areas in northeast China have obvious had advantages in terms of efficiency, which indicates that the efficiency level in northeast China is much higher than that of the other two regions, and the efficiency level in the Huang-Huai-Hai region is the lowest, which was also confirmed in the efficiency data before adjustment. In particular, as one of the three black soil areas in the world, the major grain-producing area in Northeast China has become the largest commercial grain production base in China, with grain output having exceeded 1/5 of the national total for many years. It is the “cornerstone and stabilizer” of national food security. It is not difficult to tell from Table 5 that, during the 15th Five-year Plan to the 13th Five-Year Plan period, the adjusted comprehensive technical efficiency of each region increased gradually, indicating that the resource allocation management level of the three grain production regions continued to improve during the planned period of each stage. However, from the perspective of the actual resource efficiency of grain production in the major producing areas before adjustment, the overall efficiency decline during the 11th Five-Year Plan period was mainly caused by the decrease in the efficiency of the major producing areas in northeast China and the middle and upper reaches of the Yangtze River. On the contrary, the allocation level of the major producing areas in the Huang-Huai-Hai region still rose steadily. The main reason is that during the 11th Five-Year Plan period, the Chinese government and all provinces invested heavily in the construction of grain production infrastructure in the Huang-Huai-Hai region, which improved the grain production and further optimized the internal composition of the sowing area. In particular, Henan province ranked first in terms of grain output in China for five consecutive years during this period. In addition, we found that the total factor productivity index of food production resources showed an upward trend on the whole, and its change was affected by both technological efficiency and technological progress, among which technological progress had the greater impact. 

In this paper, some research conclusions were consistent with those of other scholars. For example, in terms of external environmental impact, He et al. [20] and Zhang et al. [58] found that, among external environmental indicators, the impact direction of a disaster-affected area on relaxation variables was negatively correlated, and the other influencing factors are consistent with other scholars’ studies [20,21,24]. Additionally, the trend of the overall resource allocation efficiency after adjustment is consistent with Wang Lei et al. [25]. In the efficiency comparison of major grain-producing areas, Heilongjiang and Jilin have always been typical areas of large-scale agricultural modernization in China, and their efficiency results were verified in this paper, which is also consistent with the conclusions of scholars’ first-stage data envelopment analysis [26,58]. The innovation of this paper lies in that the selection of input–output indicators, which take into account the factors of water, soil, energy, and carbon in the process of grain production, and the more cutting-edge EBM–GML measurement tool, which was used to achieve comparative efficiency; therefore, some of the results differ from those of other agricultural productivity studies. Additionally, in some difference comparisons, it is more reasonable to weigh the overall efficiency results. In terms of the comparative analysis of different regions, this paper did not divide the study area into eastern, central, and western regions according to the previous economic belt classification standard [26,58]. Instead, the main grain-producing areas were divided into the northeast region, the Huang-Huai-Hai region, and the middle and upper reaches of the Yangtze River based on the characteristics of grain planting, physical geography, and allocative efficiency. In addition, an analysis of changes to China’s “Five-Year Plan” was considered.

There are two main inspirations from the conclusions of this paper.

First, the allocation of agricultural resources is a complex systematic project, so we should not just focus on one single target or factor when considering using food production resources more efficiently. The empirical results of this paper prove that external environmental factors and random error factors have a great impact on resource utilization efficiency in the process of grain production, and the random error is generally difficult to control. Therefore, reducing the external environmental differences in the major grain-producing interval is an effective way to achieve efficiency improvement. Combining this with the regression estimation of stochastic frontier analysis in the second stage, we firstly found that, while improving urbanization, more attention should be paid to the excessive investment in agricultural machinery and equipment. Through promoting moderate land intensification to optimize the structure of agricultural machinery and equipment, we should upgrade agricultural machinery technology and enhance scientific research strength, establishing a pattern of agricultural mechanization development with Chinese characteristics. Secondly, the positive correlation between the redundancy of chemical fertilizer application amount and economic development represents the practical dilemma of chemical fertilizer application. On the one hand, excessive fertilizer in the grain production process causes increasing production cost, waste of labor resource and arable land, and soil degradation. On the other hand, carbon emissions caused by fertilizers occupy the largest proportion of the whole agricultural system. Improving fertilizer efficiency is an effective way to achieve a comprehensive carbon peak and carbon neutrality at an early date. In this regard, it is suggested that we further carry out binding measures related to chemical fertilizer reduction. We should realize chemical fertilizer reduction and efficiency improvement through technical means such as using organic fertilizer, applying fertilizer in accordance with soil conditions, targeted fertilization, mechanized deep fertilization, and soil structure improvement. Thirdly, in implementing agricultural financial support policies, more attention should be paid to the investment in various aspects such as the project and subsidy to promote the improvement of the efficiency of fertilizer and irrigation water. Finally, we should steadily promote the construction of high-standard farmland and expand the reconstruction of farmland infrastructure. Irrigation and water conservancy facilities can help to effectively avoid the impact of natural disasters on agriculture, especially in areas where flood and drought disasters have significantly reduced production. A monitoring network of agricultural meteorological conditions and natural disaster system compatible with the construction of high-standard farmland should be built. The intelligent application capacity of agricultural meteorological observation data will be realized using big data, cloud computing, and Internet of Things technologies.

Secondly, the characteristics of resource use efficiency in major grain-producing areas in China are obviously different, so each region should further adjust its efficiency based on its potential for improvement. With lower pure technical efficiency caused by inefficient utilization of resources, in areas such as Jiangsu, Hebei, Henan, Shandong, and Anhui, we can improve management and technology as a starting point and realize a greater resource allocation of grain production. These provinces should adopt better farming technologies and crop strains, strengthen the management and innovation of food production, introduce advanced management ideas and methods, and establish a new system so as to ensure the sustainable and efficient growth of food production resource utilization. For regions with rather low scale efficiency, such as Jiangxi, Liaoning, and Neimenggu, the allocation efficiency can be improved by tapping the production scale, including promoting grain cultivation on a moderate scale in line with intensive operation, accelerating the construction of high-standard farmland, and guiding grain from small-scale farmers’ scattered cultivation to cooperative enterprises’ large-scale cultivation. This will help turn the disorderly and extensive production model into a green and efficient development model. In view of the low pure technical efficiency and scale efficiency in Hubei province, we should adopt new technology and strengthen management with comprehensive management measures, according to local conditions, to promote grain production resource allocation.

The limitation of this paper is that, due to the lack of better statistical data, the indicators can only be selected from national macro statistical data and some scholars’ calculation data. In particular, only six types of indicators were used to calculate the carbon emission data, which did not take into account the carbon emission in the processing stage after grain harvesting, as well as the role of agricultural carbon sink and land organic matter protection. In particular, carbon emission data only adopted six types of indicators in its calculation, and the carbon emission before and after food production was not taken into account. However, this did not affect the overall conclusion of this paper. The contribution of this paper lies in proving that the external environmental and random error factors of grain production in the process of resource utilization efficiency have a large impact. As such, it points to steps that managers should take to reduce the external environmental differences between the major grain-producing areas as a way of increasing efficiency. Thus, our adjustment of the resource allocation efficiency of grain production for regions may provide a theoretical reference. Future studies in this area will attempt to evaluate the impact of partial technological progress and resource allocation interaction on green grain production and provide a theoretical basis for the low-carbon transformation of food production. In addition, the impact of carbon sink and soil improvement in the process of food production should be considered in the following research on the resource allocation of food production. Meanwhile, food waste in the process of food processing and consumption in the future also needs more attention.

## 5. Conclusions

With carbon emission constraints in place, taking water–soil–energy–carbon correlation as an indicator, this paper conducted static and dynamic analyses and research on the utilization efficiency of grain production resources in 13 provinces in the major grain-producing areas in China from 2000 to 2019, based on the three-stage super-efficiency EBM–GML model under the global common frontier boundary. The paper drew the following conclusions:(1)Applying the traditional envelope analysis model, the weighted mean value of the utilization efficiency of grain production resources in major producing areas from 2000 to 2019 was 0.733, with great differences between the different regions. Jilin, Jiangxi, and Heilongjiang were the top three regions in terms of efficiency level. During the study period, the utilization efficiency of grain production resources in the major producing areas was relatively high, and the annual growth rate was 0.49%, but there is still more than 20% room for improvement.(2)After excluding external environmental and random factors, the weighted mean value of grain production resource utilization efficiency in the major producing areas decreased to 0.639, and the efficiency and ranking of each province changed greatly. External factors inhibited pure technical efficiency while expanding scale efficiency, and finally the comprehensive technical efficiency was exaggerated with the scale efficiency in place. Meanwhile, the direction and intensity of the influence of external factors on the utilization efficiency of grain production resources were obviously different in each region, which proves that the influence of external factors on the efficiency is not always positive.(3)In terms of the spatiotemporal characteristics of the utilization efficiency of grain production resources in the major producing areas after adjustment, the allocation efficiency increased from 0.528 to 0.761 during the study period, with an annual growth rate of 1.94%. The improvement in the technical level of grain production resource allocation was not as fast as that of production scale. Our research found that the utilization efficiency of grain production resources in northeast China was much higher than that of the other two regions after 2005, and the major grain-producing areas in northeast China had obvious advantages in terms of the allocation and management level of grain production resources.(4)The change index of the total factor productivity of grain production resources in the major producing areas showed an upward trend on the whole, and the change was basically consistent with the changing trend of technological progress. This change was more influenced by technological advances. After excluding external factors, the total factor productivity of resources in northeast China showed the fastest growth. Technological progress had an obvious effect on the growth of total factor productivity of grain production resources in Heilongjiang, Jiangsu, Hubei, and Hunan provinces.

## Figures and Tables

**Figure 1 ijerph-19-07746-f001:**
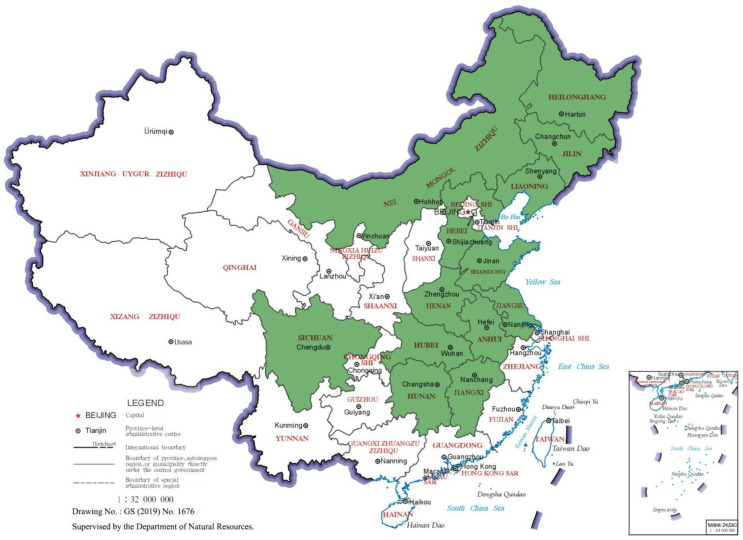
Distribution diagram of major grain-producing areas in China. The green part of the map shows the distribution of major grain-producing provinces in China.

**Figure 2 ijerph-19-07746-f002:**
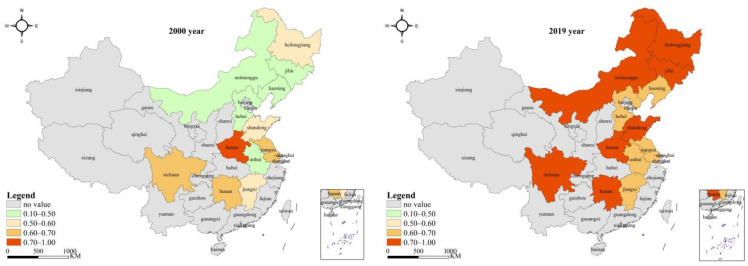
Comparison of the comprehensive technical efficiency of grain production resource allocation in the major grain-producing areas in China in 2000 and 2019 after adjustment.

**Figure 3 ijerph-19-07746-f003:**
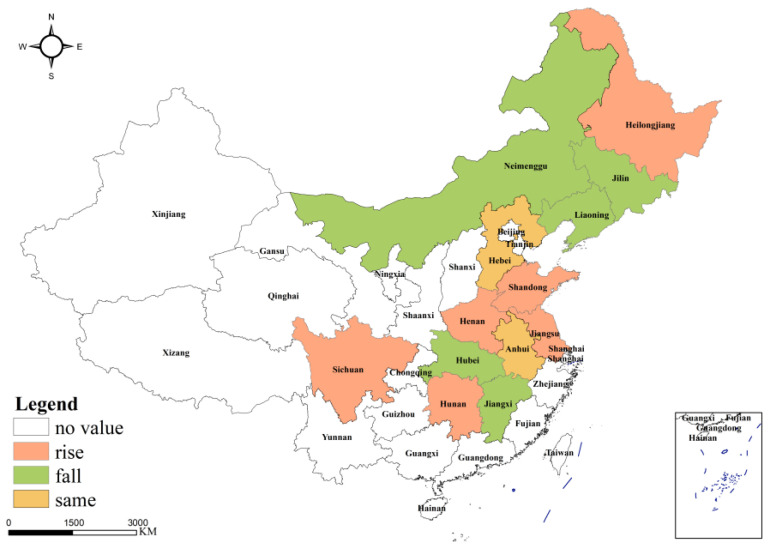
The ranking changes in grain production resource allocation efficiency (comprehensive technical efficiency) in the main grain-producing areas in China after adjustment.

**Figure 4 ijerph-19-07746-f004:**
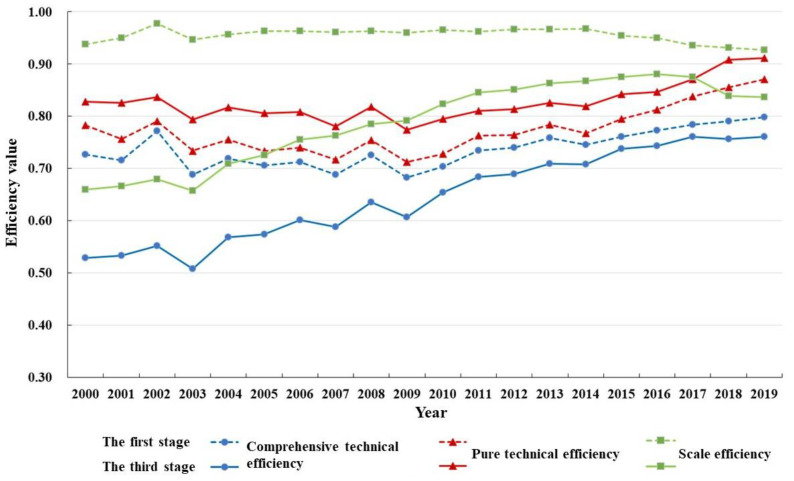
Evolving trend of grain production resource allocation efficiency in the major grain-producing areas in China from 2000 to 2019.

**Figure 5 ijerph-19-07746-f005:**
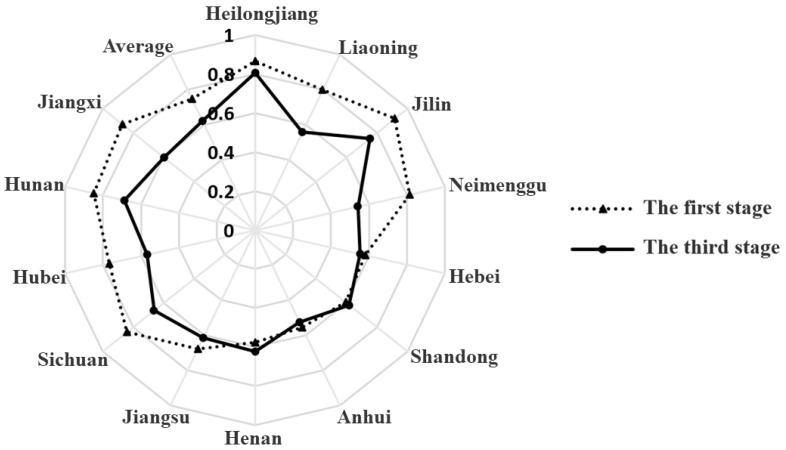
Comparison of averages of comprehensive technical efficiency of grain production resource allocation in different regions before and after adjustment.

**Figure 6 ijerph-19-07746-f006:**
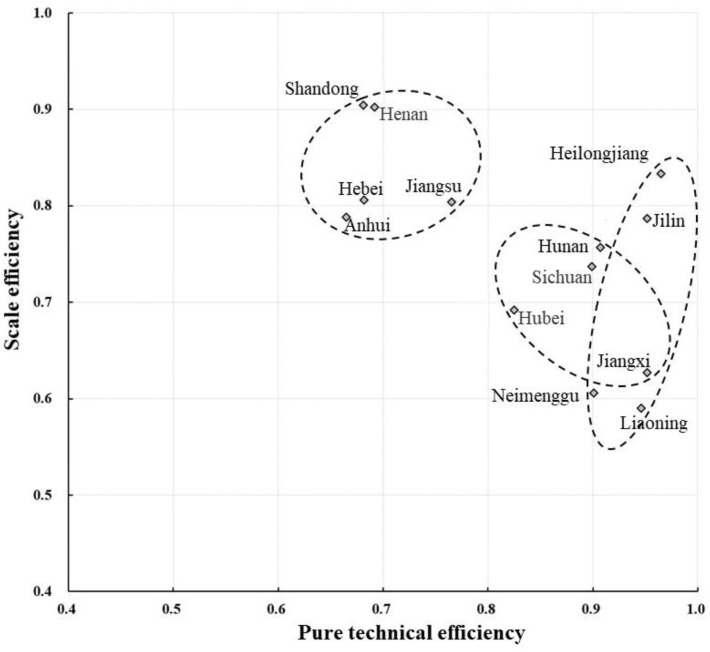
Scatter diagram of pure technical efficiency and scale efficiency after adjustment.

**Figure 7 ijerph-19-07746-f007:**
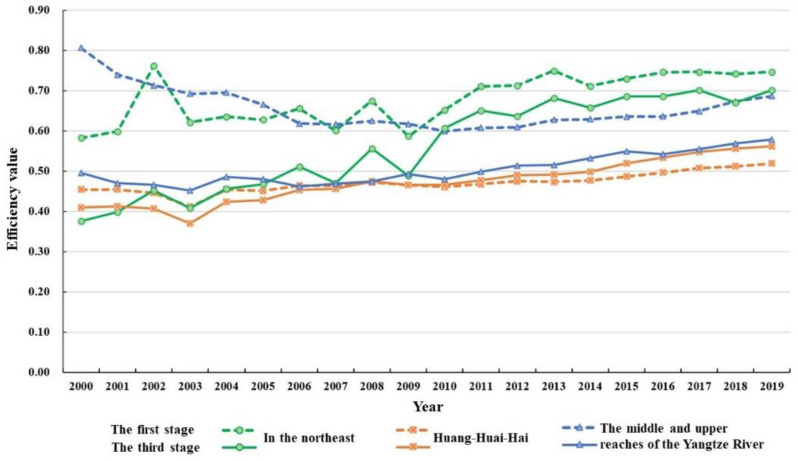
Evolving trend of grain production resource allocation efficiency in the three major regions from 2000 to 2019.

**Figure 8 ijerph-19-07746-f008:**
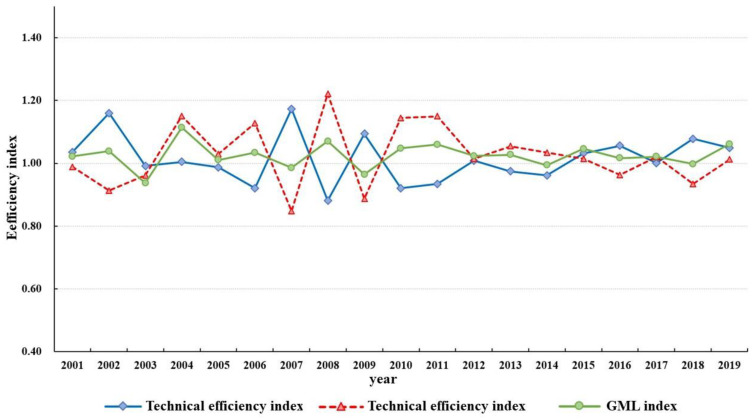
The change index and decomposition index trend of total factor productivity of grain production resources in major grain-producing areas from 2000 to 2019 after adjustment.

**Table 1 ijerph-19-07746-t001:** The comprehensive technical efficiency, pure technical efficiency, and scale efficiency of grain production resources allocation in the major grain-producing areas of China from 2000 to 2019 in the first stage.

Region	Comprehensive Technical Efficiency		Pure Technical Efficiency		Scale Efficiency	
	Efficiency Value	Rank	Efficiency Value	Rank	Efficiency Value	Rank
Heilongjiang	0.866	3	0.894	3	0.966	9
Liaoning	0.798	7	0.913	2	0.877	12
Jilin	0.917	1	0.921	1	0.995	1
Neimenggu	0.813	6	0.871	5	0.937	11
Hebei	0.581	11	0.586	12	0.991	3
Shandong	0.594	10	0.621	11	0.959	10
Anhui	0.556	13	0.569	13	0.978	5
Henan	0.576	12	0.669	10	0.875	13
Jiangsu	0.677	9	0.694	9	0.977	6
Sichuan	0.841	5	0.857	6	0.983	4
Hubei	0.767	8	0.789	8	0.974	8
Hunan	0.849	4	0.853	7	0.994	2
Jiangxi	0.871	2	0.892	4	0.976	7
Max	0.917		0.921		0.995	
Mini	0.556		0.569		0.875	
Weighted average	0.733		0.767		0.957	

Note: The three efficiency values of each region in the table are the average values from 2000 to 2019. The overall efficiency is a weighted average of the total output of each region in the major producing areas.

**Table 2 ijerph-19-07746-t002:** Regression results of the second stage SFA.

Dependent Variable				
Independent Variable	Grain Water Footprint Relaxation Variables	Grain Sown Area Relaxation Variable	Fertilizer Relaxation Variable	Total Dynamic Relaxation Variable of Machinery
Level of urbanization	−0.305 (−0.77)	−5.33 × 10^2^ (−0.65)	−0.725 *** (−2.80)	5.08 × 10^3^ *** (8.83)
Disaster degree	−7.21 × 10^−3^ *** (−5.43)	−0.167 *** (−9.32)	−2.28 × 10^−3^ ** (−2.39)	2.42 × 10^−3^ (6.52 × 10^−2^)
Level of economic development	−7.23 × 10^−4^ *** (−4.46)	−1.57 × 10^−2^ *** (−7.49)	5.71 × 10^−4^ *** (3.84)	−2.50 × 10^−2^ *** (−5.51)
Resources endowment	1.73 × 10^−3^ (0.95)	8.34 × 10^−2^ *** (3.91)	−4.80 × 10^−3^ *** (−3.73)	−7.65 × 10^−3^ (−0.12)
Financial support for agriculture	5.24 × 10^−2^ *** (4.46)	0.241 (1.48)	2.44 × 10^−2^ ** (2.57)	−0.136 (−0.44)
Sigma-squared	1.87 × 10^4^ *** (1.35 × 10^4^)	8.34 × 10^5^ *** (7.21 × 10^5^)	3.05 × 10^3^ *** (5.88)	3.63 × 10^6^ *** (3.58 × 10^6^)
Gamma	0.984 *** (6.89 × 10^2^)	0.937 *** (1.72 × 10^2^)	0.948 *** (77.4)	0.926 *** (131)
Log likelihood	−1090	−1750	−1010	−1920
LR	735	376	452	390

Note: The *t* values of the corresponding coefficients are in parentheses. *** and ** are the statistical significance levels of the corresponding system at the 0.01 and 0.05 levels, respectively.

**Table 3 ijerph-19-07746-t003:** The comprehensive technical efficiency, pure technical efficiency, and scale efficiency of grain production resource allocation in major grain-producing areas of China from 2000 to 2019 in the third stage.

Region	Comprehensive Technical Efficiency		Pure Technical Efficiency		Scale Efficiency	
	Efficiency Value	Rank	Efficiency Value	Rank	Efficiency Value	Rank
Heilongjiang	0.807	1 (rise)	0.965	1 (rise)	0.833	3 (rise)
Liaoning	0.557	10 (fall)	0.946	4 (fall)	0.590	13 (fall)
Jilin	0.753	2 (fall)	0.952	2 (fall)	0.787	7 (fall)
Neimenggu	0.541	12 (fall)	0.901	6 (fall)	0.606	12 (fall)
Hebei	0.553	11 (same)	0.682	11 (rise)	0.806	4 (fall)
Shandong	0.618	6 (rise)	0.681	12 (fall)	0.904	1 (rise)
Anhui	0.525	13 (same)	0.665	13 (same)	0.788	6 (fall)
Henan	0.622	5 (rise)	0.692	10 (same)	0.902	2 (rise)
Jiangsu	0.615	7 (rise)	0.765	9 (same)	0.804	5 (rise)
Sichuan	0.663	4 (rise)	0.899	7 (fall)	0.737	9 (fall)
Hubei	0.569	9 (fall)	0.825	8 (same)	0.692	10 (fall)
Hunan	0.686	3 (rise)	0.907	5 (rise)	0.757	8 (fall)
Jiangxi	0.596	8 (fall)	0.952	3 (rise)	0.627	11 (fall)
Max	0.807		0.965		0.904	
Mini	0.525		0.665		0.590	
Weighted average	0.639		0.822		0.785	

Note: The three efficiency values of each region in the table are the average values from 2000 to 2019. The overall efficiency is a weighted average of the total output of each region in the major producing areas.

**Table 4 ijerph-19-07746-t004:** Grain production resource allocation efficiency in the major grain-producing areas in China from 2000 to 2019.

Years	The Grain Production Resource Allocation Efficiency in the Major Grain-Producing Areas in China From 2000 to 2019 (In the First and Third Stages)			The First and Third Stages of the Three Regional Comprehensive Technical Efficiencies		
	Comprehensive Technical Efficiency	Pure Technical Efficiency	Scale Efficiency	In the Northeast	Huang-Huai-Hai	The Middle and Upper Reaches of the Yangtze River
2000	0.727 (0.528)	0.782 (0.828)	0.938 (0.660)	0.583 (0.375)	0.454 (0.409)	0.807 (0.496)
2001	0.715 (0.533)	0.756 (0.826)	0.950 (0.666)	0.598 (0.399)	0.454 (0.413)	0.740 (0.470)
2002	0.772 (0.551)	0.790 (0.837)	0.977 (0.679)	0.762 (0.452)	0.445 (0.407)	0.713 (0.465)
2003	0.688 (0.508)	0.734 (0.794)	0.946 (0.657)	0.621 (0.409)	0.412 (0.370)	0.692 (0.452)
2004	0.719 (0.568)	0.755 (0.817)	0.957 (0.708)	0.635 (0.455)	0.454 (0.424)	0.695 (0.486)
2005	0.705 (0.574)	0.734 (0.805)	0.963 (0.725)	0.628 (0.468)	0.451 (0.428)	0.665 (0.480)
2006	0.712 (0.601)	0.740 (0.807)	0.963 (0.755)	0.656 (0.511)	0.464 (0.452)	0.618 (0.462)
2007	0.688 (0.587)	0.717 (0.780)	0.960 (0.763)	0.600 (0.470)	0.464 (0.456)	0.616 (0.469)
2008	0.726 (0.635)	0.754 (0.818)	0.963 (0.785)	0.675 (0.556)	0.472 (0.474)	0.625 (0.474)
2009	0.683 (0.607)	0.712 (0.774)	0.959 (0.791)	0.586 (0.488)	0.466 (0.465)	0.617 (0.493)
2010	0.703 (0.654)	0.728 (0.794)	0.965 (0.824)	0.652 (0.607)	0.460 (0.466)	0.599 (0.480)
2011	0.734 (0.684)	0.762 (0.810)	0.962 (0.845)	0.710 (0.650)	0.468 (0.477)	0.608 (0.498)
2012	0.739 (0.689)	0.763 (0.813)	0.966 (0.851)	0.712 (0.636)	0.475 (0.490)	0.609 (0.513)
2013	0.759 (0.709)	0.784 (0.825)	0.966 (0.862)	0.749 (0.681)	0.473 (0.491)	0.627 (0.515)
2014	0.745 (0.708)	0.767 (0.818)	0.968 (0.867)	0.711 (0.658)	0.477 (0.498)	0.629 (0.532)
2015	0.760 (0.737)	0.795 (0.842)	0.954 (0.875)	0.730 (0.685)	0.487 (0.520)	0.635 (0.549)
2016	0.773 (0.743)	0.812 (0.846)	0.950 (0.881)	0.746 (0.686)	0.496 (0.533)	0.636 (0.543)
2017	0.784 (0.761)	0.837 (0.870)	0.936 (0.875)	0.746 (0.702)	0.508 (0.547)	0.649 (0.554)
2018	0.791 (0.756)	0.855 (0.907)	0.931 (0.839)	0.742 (0.670)	0.512 (0.556)	0.673 (0.568)
2019	0.798 (0.761)	0.871 (0.911)	0.927 (0.836)	0.746 (0.701)	0.519 (0.561)	0.687 (0.578)
Weighted average	0.733 (0.63)	0.767 (0.822)	0.957 (0.785)	0.679 (0.563)	0.471 (0.472)	0.657 (0.504)

Note: In the table, the values of comprehensive technical efficiency, pure technical efficiency, and scale efficiency are the weighted average values of grain output in the total output of major grain-producing areas, and the values in brackets are the efficiency values of the third stage after adjustment.

**Table 5 ijerph-19-07746-t005:** Regional grain production resource allocation efficiency during five-year planning periods.

Period	In the Northeast	Huang-Huai-Hai	The Middle and Upper Reaches of the Yangtze River	Major Grain Producing Areas
The 10th Five-year Plan	0.649 (0.436)	0.443 (0.408)	0.701 (0.471)	0.720 (0.547)
The 11th Five-Year Plan	0.634 (0.526)	0.465 (0.463)	0.615 (0.476)	0.702 (0.617)
The 12th Five-Year Plan	0.722 (0.662)	0.476 (0.495)	0.622 (0.521)	0.747 (0.705)
The 13th Five-Year Plan	0.745 (0.690)	0.509 (0.549)	0.661 (0.561)	0.787 (0.755)

Note: Due to insufficient research years, the 13th Five-Year Plan is the average value from 2016 to 2019. The efficiency value in the table is the weighted average value of grain output in the total output of major producing areas, and the adjusted comprehensive technical efficiency value is in parentheses.

**Table 6 ijerph-19-07746-t006:** Dynamic change index of total factor productivity of grain production resources in the major producing areas from 2000 to 2019.

Region	Technical Efficiency Change Index	Technological Progress Change Index	Pure Technical Efficiency Change Index	Scale Efficiency Change Index	GML (TFP) Change Index
Heilongjiang	1.014 (0.997)	1.013 (1.041)	1.000 (0.996)	1.015 (1.004)	1.025 (1.037)
Liaoning	1.032 (1.021)	1.000 (1.022)	1.000 (1.000)	1.033 (1.021)	1.023 (1.038)
Jilin	1.020 (1.028)	0.993 (1.011)	1.020 (1.005)	1.000 (1.022)	1.010 (1.039)
Neimenggu	1.012 (1.045)	1.024 (1.037)	1.005 (1.002)	1.010 (1.044)	1.023 (1.047)
Hebei	1.019 (1.009)	1.001 (1.019)	1.021 (1.016)	0.999 (0.994)	1.017 (1.024)
Shandong	1.015 (1.027)	0.999 (1.018)	1.011 (1.008)	1.027 (1.025)	1.009 (1.017)
Anhui	1.016 (1.006)	0.998 (1.023)	1.018 (1.007)	1.000 (0.997)	1.008 (1.024)
Henan	1.016 (1.025)	0.998 (1.014)	1.001 (1.010)	1.014 (1.028)	1.007 (1.022)
Jiangsu	1.025 (0.999)	1.022 (1.026)	1.022 (1.011)	1.004 (0.991)	1.001 (1.009)
Sichuan	1.016 (1.007)	1.006 (1.044)	1.008 (1.007)	1.008 (0.998)	0.994 (1.011)
Hubei	0.993 (0.990)	0.996 (1.023)	0.992 (0.992)	1.004 (1.001)	0.982 (1.006)
Hunan	1.008 (0.995)	0.994 (1.025)	1.010 (1.008)	1.001 (0.987)	0.989 (1.005)
Jiangxi	1.002 (1.003)	1.006 (1.024)	0.991 (0.997)	1.012 (1.004)	1.000 (1.015)
In the northeast	1.020 (1.023)	1.007 (1.028)	1.006 (1.001)	1.014 (1.022)	1.020 (1.040)
Huang-Huai-Hai	1.018 (1.013)	1.004 (1.020)	1.015 (1.011)	1.009 (1.007)	1.008 (1.019)
The middle and upper reaches of the Yangtze River	1.004 (0.999)	1.000 (1.029)	1.000 (1.001)	1.007 (0.998)	0.991 (1.009)
Average	1.014 (1.012)	1.004 (1.025)	1.008 (1.005)	1.010 (1.009)	1.007 (1.023)

Note: Each efficiency change index is the average from 2000 to 2019, and the adjusted efficiency change index of the third stage is in parentheses.

## Data Availability

The crop growth data and meteorological data were derived from the “study on the water demand contour map of China’s major crops (books)”, “Crop-wat Software”, the database of The United Nations Food and Agriculture Organization (https://www.fao.org/statistics/zh/, (accessed on 1 October 2021)), the China Meteorological Data Network (http://data.cma.cn/, (accessed on 15 October 2021)), provincial statistical yearbooks, and previous research (please refer to the paragraph in Section 3.2 for details). Additionally, the basic data on carbon sources were collected from provincial statistical yearbooks. In addition to the calculation of crops’ water footprint and carbon emissions, the remaining data indicators and variables are derived from the provincial statistical yearbooks, the National Bureau of Statistics database (https://data.stats.gov.cn/, (accessed on 8 January 2022)), and the “China rural statistical yearbook” (https://data.cnki.net/ (accessed on 8 January 2022)) from 2001 to 2020.

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
