# Peer review of "An Analysis of the Spatiotemporal Characteristics and Diversity of Grain Production Resource Utilization Efficiency under the Constraint of Carbon Emissions: Evidence from Major Grain-Producing Areas in China"

_ijerph, 2022, doi:10.3390/ijerph19137746_

Round 1
Reviewer 1 Report
The aim of the research was to analyze the possibilities of cereal production in northeastern China on the basis of data obtained in the years 2000-20019 depending on biotic and abiotic factors. The analysis was performed on the basis of statistical methods using available and valid models. The work is interesting and shows which elements of cereal production should be changed to maximize the yield of cereals per 1 ha.
However, the publisher has some disadvantages.
The results should be shortened and most of the content transferred to the discussion. No research hypothesis at work.
After considering the comments, the work may be published in Int. J. Environ. Res. Public Health
Author Response
Dear Professor,
we have carefully revised the manuscript according to your opinion, and your opinion has greatly improved the quality of our paper. Please see the attachment.
Looking forward to hearing from you.
Thank you and best regards.
Yours sincerely,
Hong Chen
Tel: (86)13936441436
Email: chenhong@nefu.edu.cn
College of Economics and Management, Northeast Forestry University, Harbin 150000, China

Reviewer 2 Report
The paper is really interesting, the main contribution of the paper is assessing the efficiency in agricultural production of the sub-regions in China after eliminating external factors influence. The authors used appropriate data sets and methodology to arrive at conclusion. It adds value to the existing literature on efficiency measurement.
However, there is a need for more clarity in presentation. Sometimes authors just explained the model and more emphasis is given just narrating what is there in the tables. Paper will be improved a lot, if authors explain real economic rationale of employing three stages of analysis clearly at the beginning itself.
Some tables may be converted in to maps (for example efficiency data super imposed on China geographical map) and also change in efficiency (1st and 3rd stage analysis on the map of China) can improve the readability of the paper greatly.
The literature on India is missing, India is a comparable country authors may include some literature on India to attract wider audience for the paper. Also include EU studies mentioned below.
Reddy, A. A. (2010). Disparities in agricultural productivity growth in Andhra Pradesh. The Indian Economic Journal, 58(1), 134-152.
Toma, P., Miglietta, P. P., Zurlini, G., Valente, D., & Petrosillo, I. (2017). A non-parametric bootstrap-data envelopment analysis approach for environmental policy planning and management of agricultural efficiency in EU countries. Ecological indicators, 83, 132-143.
There is need for improving the English language, there are many typos” examples are mentioned below
Line no. 121: it is regional heterogeneity
LN. No. 128-29: not clear.
LN No. 201-204 not clear.
LN No. 214-217: give main idea behind 1st stage.
LN No. 253-254254-257: have you gave weights to wheat, paddy based on their global prices? Or each crop is taken as separate output?
LN 317: and not of.
338-344: improve.
369: check.
434: check
Author Response

(The authors gave the same response as above.)

Reviewer 3 Report
Please find attached file with my comments
regards

Author Response

(The authors gave the same response as above.)
